# Ordering-Based Causal Discovery with Reinforcement Learning

## Abstract

It is a long-standing question to discover causal relations among a set of variables in many empirical sciences. Recently, Reinforcement Learning (RL) has achieved promising results in causal discovery. However, searching the space of directed graphs directly and enforcing acyclicity by implicit penalties tend to be inefficient and restrict the method to small problems. In this work, we alternatively consider searching an ordering by RL from the variable ordering space that is much smaller than that of directed graphs, which also helps avoid dealing with acyclicity. Specifically, we formulate the ordering search problem as a Markov decision process, and then use different reward designs to optimize the ordering generating model. A generated ordering is then processed using variable selection methods to obtain the final directed acyclic graph. In contrast to other causal discovery methods, our method can also utilize a pretrained model to accelerate training. We conduct experiments on both synthetic and real-world datasets, and show that the proposed method outperforms other baselines on important metrics even on large graph tasks.

## 1 Introduction

Identifying causal structure from observational data is an important but also challenging task in many applications. This problem can be formulated as that of finding a Directed Acyclic Graph (DAG) that minimizes some score function defined w.r.t. observed data. Though there exist well-studied score functions like Bayesian Information Criterion (BIC) or Minimum Description Length (MDL) (Schwarz et al., 1978; Chickering, 2002), searching over the space of DAGs is known to be NP-hard, even if each node has at most two parents (Chickering, 1996). Consequently, traditional methods mostly rely on local heuristics to perform the search, including greedy hill climbing and Greedy Equivalence Search (GES) that explore the Markov equivalence classes (Chickering, 1996).

Along with various search strategies, existing methods have also considered to reduce the search space while meeting the DAG constraint. A useful practice is to cast the causal structure learning problem as that of learning an optimal ordering of variables (Koller & Friedman, 2009). Because the ordering space is significantly smaller than the space of directed graphs and searching over ordering space can avoid the problem of dealing with acyclic constraints (Teyssier & Koller, 2005). Many algorithms are used to search for ordering such as genetic algorithm (Larranaga et al., 1996), Markov chain Monte Carlo (Friedman & Koller, 2003) and greedy local hill-climbing (Teyssier & Koller, 2005). However, these algorithms often cannot find the best ordering effectively.

Recently, with differentiable score functions, several gradient-based methods have been proposed based on a smooth characterization of acyclicity, including NOTEARS (Zheng et al., 2018) for linear causal models and several subsequent works, e.g., Yu et al. (2019); Lachapelle et al. (2020); Ng et al. (2019b;a), which use neural networks to model nonlinear causal relationships. As another attempt, Zhu et al. (2020) utilize Reinforcement Learning (RL) as a search strategy to find the best DAG from the graph space and it can be incorporated with a wide range of score functions. Unfortunately, its good performance is achieved only with around 30 variables, for at least two reasons: 1) the action space, consisting of directed graphs, is tremendous for large scale problems and hard to be explored efficiently; and 2) it has to compute scores for many non-DAGs generated during training but computing scores w.r.t. data is generally be time-consuming. It appears that the RL-based approach may not be able to achieve a close performance to other gradient-based methods that directly optimize the same (differentiable) score function for large causal discovery problems, due to its search nature.

In this work, we propose a RL-based approach for causal discovery, named Causal discovery with Ordering-based Reinforcement Learning (CORL), which combines RL with the ordering based paradigm so that we can exploit the powerful search ability of RL to search the best ordering efficiently. To achieve this, we formulate the ordering search problem as a Markov Decision Process (MDP), and then use different reward designs for RL to optimize the ordering generating model. In addition, we notice that pretrained model can be incorporated into our model to accelerate training. For a generated ordering, we prune it to the final DAG by variable selection. The proposed approach is evaluated on both synthetic and real datasets to validate its effectiveness. In particular, the proposed method can achieve a much improved performance than the previous RL-based method on both linear and non-linear data models, even outperforms NOTEARS, a gradient-based method, on 150-node linear data models, and is competitive with Causal Additive Model (CAM) on non-linear data models.

## 2 RELATED WORKS

Existing causal discovery methods roughly fall into three classes. The first class, as described in the introduction, are the so-called score-based methods. Besides the mentioned BIC/MDL scores, other score functions include the Bayesian Gaussian equivalent score (Geiger & Heckerman, 1994), the generalized score based on (conditional) independence relationship (Huang et al., 2018), and a recently proposed meta-transfer score (Bengio et al., 2020). Another class of methods such as fast causal inference and PC (Spirtes et al., 2000; Zhang, 2008) which first find causal skeleton and then decide the orientations of the edges up to the Markov equivalence class, are viewed as constraint-based. Such methods usually involve multiple independent testing problems; the testing results may have conflicts and handling them is not easy. The last class of methods relies on properly defined functional causal models, includeing Linear Non-Gaussian Acyclic Model (LiNGAM), nonlinear Additive Noise Model (ANM) (Hoyer et al., 2009; Peters et al., 2014), and the post-nonlinear causal model (Zhang & Hyvärinen, 2009). By placing certain assumptions on the class of causal functions and/or noise distributions, these methods can distinguish different DAGs in the same Markov equivalence class.

Since we particularly consider ordering-based approaches, here we present a more detailed review of such methods. Most of the ordering-based methods belong to the class of score-based methods. Besides the mentioned heuristic search algorithms, Schmidt et al. (2007) proposed L1OBS to conduct variable selection using $\ell_1$-regularization paths based on Teyssier & Koller (2005). Scanagatta et al. (2015) further proposed an ordering exploration method on the basis of an approximated score function so as to scale to thousands of variables. The Causal Additive Model (CAM) was developed by Bühlmann et al. (2014) for nonlinear data models. Some recent ordering-based methods such as sparsest permutation (Raskutti & Uhler, 2018) and greedy sparest permutation (Solus et al., 2017) can guarantee consistency of Markov equivalence class, based on some conditional independence relations and certain assumptions like faithfulness. A variant of greedy sparest permutation was also proposed in Bernstein et al. (2020) for the setting with latent variables. In the present work, we mainly work on identifiable cases which may have different assumptions from theirs.

In addition, several exact algorithms such as dynamic programming (Silander & Myllymäki, 2006; Perrier et al., 2008) and integer or linear programming (Jaakkola et al., 2010; Cussens, 2011; Bartlett & Cussens, 2017) are used to solve causal discovery problem. However, these exact algorithms usually work on small graphs efficiently (De Campos & Ji, 2011), and in order to handle larger problems with hundreds of variables, they need to incorporate heuristics search (Xiang & Kim, 2013) or limit the maximum number of parents.

Recently, RL has been used to tackle several combinatorial problems such as maximum cut and the traveling salesman problem, due to their relatively simple reward mechanisms (Khalil et al., 2017). In combination with the encoder-decoder based pointer networks (Vinyals et al., 2015), Bello et al. (2016) showed that RL can have a better generalization even when the optimal solutions are used as labeled data in a supervised way. Kool et al. (2019) further used an attention based encoder-decoder model for an improved performance. These works aim to learn a policy as a combinatorial solver based on the same structure of a particular type of combinatorial problems. However, various causal discovery tasks generally have different causal relationships, data types, graph structures, etc, and moreover, are typically off-line with focus on a causal graph. As such, we use RL as a search strategy, similar to Zhu et al. (2020); Zoph & Le (2017); nevertheless, a pretrained model or a policy can offer a good starting point to speed up training, as shown in our evaluation results (Figure 3).

## 3 BACKGROUND

### 3.1 CAUSAL STRUCTURE LEARNING

Let $\mathcal{G} = (d, V, E)$ denotes a DAG, with $d$ the number of nodes, $V = \{v_1, \ldots, v_d\}$ the set of nodes, and $E = \{(v_i, v_j)|i, j = 1, \ldots, d\}$ the set of directed edges from $v_i$ to $v_j$. Each node $v_j$ is associated with a random variable $X_j$. The probability model associated with $\mathcal{G}$ factorizes as $p(X_1, \ldots, X_d) = \prod_{j=1}^{d} p(X_j|\mathrm{Pa}(X_j))$, where $p(X_j|\mathrm{Pa}(X_j))$ is the conditional probability distribution for $X_j$ given its parents $\mathrm{Pa}(X_j) := \{X_k|(v_k, v_j) \in E\}$. We assume that the observed data $\mathbf{x}_j$ is obtained by the Structural Equation Model (SEM) with addtive noise: $X_j := f_{\theta_j}(\mathrm{Pa}(X_j)) + \epsilon_j, j = 1, \ldots, d$, where $f_{\theta_j}$ parameterized by $\theta_j$ is used to represent the functional relationship between $X_j$ and its parents, and $\epsilon_j$'s denote jointly independent additive noise variables. We assume causal minimality, which is equivalent to that each $f_j$ is not a constant for any $X_k \in \mathrm{Pa}(X_j)$ in this SEM (Peters et al., 2014).

Given a sample $\mathbf{X} = [\mathbf{x}_1, \ldots, \mathbf{x}_d]$, where $\mathbf{X} \in \mathbb{R}^{m \times d}$ and $\mathbf{x}_j$ is a vector of $m$ observations for random variables $X_j$. Our goal in this paper is to find the DAG $\mathcal{G}$ that maximizes the BIC score (or other well studied scores), defined as

$$\mathrm{Score}_{\mathrm{BIC}}(\mathcal{G}) = \sum_{j=1}^{d} \left[ \sum_{k=1}^{m} \log p(X_j^k|\mathrm{Pa}(X_j^k), \theta_j) - \frac{|\theta_j|}{2} \log m \right], \tag{1}$$

where $|\theta_j|$ is the number of free parameters in $p(X_j^k|\mathrm{Pa}(X_j^k), \theta_j)$. For linear causal relationships, $|\theta_j| = |\mathrm{Pa}(X_j^k)|$ the number of parents, up to some constant factor.

The problem of finding a directed graph that satisfies the ayclicity constraint can be cast as that of finding an variable ordering (Teyssier & Koller, 2005; Schmidt et al., 2007). The score of an ordering is usually defined as the score of the best DAG that is consistent with the given ordering. Specifically, let $\Pi$ denote an ordering of the nodes in $V$, where the length of the ordering $|\Pi| = |V|$ and $\Pi$ is indexed from 1. If node $v_j \in V$ lies in the $p$-th position, then $\Pi(p) = v_j$. Notation $\Pi_{\prec v_j}$ denotes the set of nodes in $V$ that precede node $v_j$ in $\Pi$. One can establish a canonical correspondence between an ordering $\Pi$ and a fully-connected DAG $\mathcal{G}^{\Pi}$; an example with four nodes is presented in Figure 1. For a given DAG $\mathcal{G}$, it can be consistent with more than one orderings and the set of these orderings is given by

$$\Phi(\Pi) := \{\Pi : \text{the fully-connected DAG } \mathcal{G}^{\Pi} \text{ is a super-DAG of } \mathcal{G}\}, \tag{2}$$

where a super-DAG of $\mathcal{G}$ is a DAG whose edge set is a superset of that of $\mathcal{G}$. The score of an ordering is usually defined as the score of the best DAG that is consistent with the given ordering (Teyssier & Koller, 2005; Peters et al., 2014). We provide a formal description in Proposition 1 to show that it is possible to find the correct ordering with high probability in the large sample limit. Therefore, the search for the true DAG $\mathcal{G}^*$ can be decomposed to two phases: finding the correct ordering and performing variable selection (feature selection); the latter is to find the optimal DAG that is consistent with the ordering found in the first step.

$\Pi := \{v_2, v_4, v_1, v_3\}$

Figure 1: An example of the correspondence between an ordering (down) and a fully-connected DAG (top).

**Proposition 1.** *Suppose that an identifiable SEM with true causal DAG $\mathcal{G}^*$ on $X = \{X_j\}_{j=1}^{d}$ induces distribution $P(X)$. Let $\mathcal{G}^{\Pi}$ be the fully-connected DAG that corresponds to an ordering $\Pi$. If there is an SEM with $\mathcal{G}_{\Pi}$ inducing the same distribution $P(X)$, then $\mathcal{G}^{\Pi}$ must be a super-graph of $\mathcal{G}^*$, i.e., every edge in $\mathcal{G}^*$ is covered in $\mathcal{G}^{\Pi}$.*

*Proof.* The SEM with $\mathcal{G}^{\Pi}$ may not be causally minimal but can be reduced to an SEM satisfying the causal minimality condition (Peters et al., 2014). Let $\tilde{\mathcal{G}}^{\Pi}$ denotes the causal graph in the reduced SEM with the same distribution $P(X)$. Since we have assumed that original SEM is identifiable, i.e., the distribution $P(X)$ corresponds to a unique true graph, $\tilde{\mathcal{G}}^{\Pi}$ is then identical to $\mathcal{G}^*$. The proof is complete by noticing that $\mathcal{G}^{\Pi}$ is a super-graph of $\tilde{\mathcal{G}}^{\Pi}$. □

### 3.2 Reinforcement Learning

Standard RL is usually formulated as an MDP over the environment state $s \in \mathcal{S}$ and agent action $a \in \mathcal{A}$, under an (unknown) environmental dynamics defined by a transition probability $\mathcal{T}(s'|s, a)$. We use $\pi_\phi(a|s)$ to denote the policy, parameterized by $\phi$, which outputs a discrete (or continuous) distribution used to select an action from action space $\mathcal{A}$ based on state $s$. For episodic tasks, a trajectory $\tau = \{s_t, a_t\}_{t=0}^T$, where $T$ is the finite time horizon, can be collected by executing the policy repeatedly. In many cases, an immediate reward $r(s, a)$ can be received when agent executes an action. The objective of RL is to learn a policy which can maximize the expected cumulative reward along a trajectory, i.e., $J(\phi) = \mathbb{E}_{\pi_\phi}[R_0]$ with $R_0 = \sum_{t=0}^T \gamma^t r_t(s_t, a_t)$ and $\gamma \in (0, 1]$ being a discount factor. For some scenarios, the reward is only earned at the terminal time (also called episodic reward), and $J(\phi) = \mathbb{E}_{\pi_\phi}[R(\tau)]$ with $R(\tau) = r_T(s_T, a_T)$.

## 4 Method

In this section, we first introduce how to model the ordering search problem as an MDP, then we show how to use RL to find the optimal ordering, and we introduce how to process the searched ordering to obtain the final DAG, finally we provide a discussion on computational complexity.

### 4.1 Ordering Search as a Markov Decision Process

We can regard the variable ordering search problem as a muti-stage decision problem with a variable selected at each decision step. We sort the selected variables according to decision steps to obtain a variable ordering, which is defined as the searched ordering. The decision-making process is Markovian, and its elements in the problem can be defined as follows:

**State** One can directly take the sample data $\mathbf{x}_j$ of each variable $X_j$ as a state $s_j$. However, preliminary experiments show that it is difficult for feed-forward neural network models to capture the underlying causal relationships directly using observed data as states, and the data pre-processed by a module conventionally named encoder is helpful to finding the better ordering, see Appendix A.1. The encoder module embeds each $\mathbf{x}_j$ to $s_j$ and all the embedded states constitute the space $\mathcal{S} := \{s_1, \ldots, s_d\}$. Adding initial state $s_0$ to the space constitutes the complete state space $\hat{\mathcal{S}} := \mathcal{S} \cup s_0$. We use $\hat{s}_t$ to denote the state encountered at the $t$-th decision step.

**Action** We select an action (variable) from the action space constituted by all the variables at each decision step, and the space size is equal to the number of variables, $|\mathcal{A}| = d$. Compared to the previous RL-based method searching from the graph space with size $\mathcal{O}(2^{d \times d})$ (Zhu et al., 2020), the search space is smaller. Note that according to the definition of ordering, the first selected variable is the source node and the last selected node is the sink node.

**State transition** The specified state transition is related to the action selected at the current decision step. Specifically, if the selected variable is $v_j$ at the $t$-th decision step, then the state is transferred to the state $s_j \in \mathcal{S}$ which is the $j$-th output from Transformer encoder, i.e., $\hat{s}_{t+1} = s_j$.

**Reward** As we described in Section 3.1, only the variables that have been selected in previous decision steps can be the potential parents of the currently selected variable. Hence, we can design the rewards in the following cases: *dense reward* and *episodic reward*. For *dense reward* case, we can exploit the decomposability of the score function (BIC score) to calculate an immediate reward for the current decision step based on the potential parent variables (that have been selected), i.e., if $v_j$ is selected at time step $t$, the immediate reward is calculated by

$$r_t = \sum_{k=1}^m \log p(X_j^k | U(X_j), \theta_j) - \frac{|\theta_j|}{2} \log m, \tag{3}$$

where $U(X_j)$ denotes the potential parent variable set of $X_j$ and the set consists of the variables associated with the nodes in $\Pi_{\prec v_j}$. For *episodic reward* case, we directly calculate a score as an episodic reward for a complete variable ordering regardless of whether the scoring function is decomposable or not, i.e.,

$$R(\tau) = r_T(\hat{s}, a) = \text{Score}_{\text{BIC}}(\mathcal{G}^\Pi) \tag{4}$$

where $\text{Score}_{\text{BIC}}$ has been defined in Equation (1), with the set $\text{Pa}(X_j)$ of each variable $X_j$ replaced by the potential parent variable set $U(X_j)$ here.

## 4.2 Implementation and Optimization with Reinforcement Learning

We describe the neural network architectures implemented in our method, which consists of an encoder and a decoder as shown in Figure 2. Here we briefly describe the model architectures and leave details regarding model parameters to Appendix A.

**Encoder** $f_{\phi_e}^{\text{enc}} : \tilde{\mathbf{X}} \mapsto \mathcal{S}$ is used to map the observed data to the embedding space $\mathcal{S} = \{s_1, \dots, s_d\}$. For sample efficiency, we follow Zhu et al. (2020) to randomly draw $n$ samples from the dataset $\mathbf{X}$ to construct $\tilde{\mathbf{X}} \in \mathbb{R}^{n \times d}$ at each episode and use $\tilde{\mathbf{X}}$ instead of $\mathbf{X}$. We also set the embedding $s_j$ to be in the same dimension, i.e., $s_j \in \mathbb{R}^n$. For encoder choice, we conduct an empirical comparison among several possible structures such as a self-attention based encoder (Vaswani et al., 2017) and an LSTM structure. Empirically, we confirm that the self-attention based encoder in the Transformer structure performs the best, which is also used in Zhu et al. (2020). Please find more details in Appendix A.1.

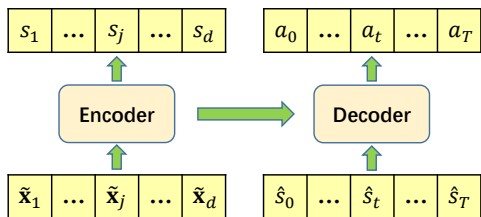

Figure 2: Illustration of the policy model. The encoder embeds the observed data $\mathbf{x}_j$ into the state $s_j$. An action $a_t$ can be selected by the decoder according to the given state $\hat{s}_t$ at each time step $t$.

**Decoder** $f_{\phi_d}^{\text{dec}} : \hat{\mathcal{S}} \mapsto \mathcal{A}$ maps the state space $\hat{\mathcal{S}}$ to the action space $\mathcal{A}$. Among several decoder choices (please see also Appendix A.1 for empirical comparison), we specifically pick an LSTM based structure that proves effective in our experiments. Although the initial state is usually generated randomly, we pick it as $s_0 = \frac{1}{d} \sum_{i=1}^{d} s_i$, considering that the source node is fixed in correct ordering. We then restrict each node only be selected once by masking the selected nodes so as to generate a valid variable ordering (Vinyals et al., 2015).

**Optimization** The optimization objective is to learn a policy which maximizes $J(\phi) = \mathbb{E}_{\pi_\phi}[R]$, where $\pi_\phi$ denotes the policy model parameterized by the paprameters $\{\phi_e, \phi_d\}$ of encoder $f^{\text{enc}}$ and decoder $f^{\text{dec}}$. Based on the above definition, policy gradient (Sutton & Barto, 2018) is used to optimize the ordering generation model parameters. For the *dense reward* case, policy gradient can be written as $\nabla J(\phi) = \mathbb{E}_{\pi_\phi} \left[ \sum_{t=0}^{T} R_t \nabla_\phi \log \pi_\phi (a_t | \hat{s}_t) \right]$, where $R_t = \sum_{l=0}^{T-t} \gamma^l r_{t+l}$ denotes the return at time step $t$. We denote the algorithm using this reward design as CORL-1. For the *episodic reward* case, we have the following policy gradient $\nabla J(\phi) = \mathbb{E}_{\pi_\phi} \left[ R(\tau) \sum_{t=0}^{T} \nabla_\phi \log \pi_\phi (a_t | \hat{s}_t) \right]$, the algorithm using this reward design is denoted as CORL-2. Using a parametric baseline to estimate the expected score typically improves learning (Konda & Tsitsiklis, 2000). Therefore, we introduce a critic network $V_{\phi_v}(\hat{s}_t)$ parameterized by $\phi_v$, which learns the expected return given a state $\hat{s}_t$. It is trained with stochastic gradient descent using Adam optimizer on a mean squared error objective between its predicted value and the actual return. The details about the parameters of critic network are described in Appendix A.2.

Inspired by the benefits of pretrained models in other tasks (Hinton & Salakhutdinov, 2012), we also consider to incorporate it to our method to accelerate training. Usually, one can obtain some observed data with known causal structure or correct ordering, e.g., by simulation or real data with labeled graph. Hence, we pretrain a policy model with such data in a supervised way.

So far we presented CORL in a general manner without specifying explicitly which distribution family is during the evaluation of rewards. In principle, any distribution family could be employed as long as its log-likelihood can be computed and no differentiability is required. However, it is not always clear whether the maximization of the accumulated reward recovers the correct ordering. It will depend on both the modelling choice of reward and the underlying SEM; in fact, if the causal relationships fall into the chosen model functions and a right distribution family is assumed, then given infinite samples the optimal accumulated reward, corresponding to the negative log-likelihood, must be achieved by a super-DAG of the underlying graph according to Proposition 1. In practice, we

---

**Algorithm 1** CORL.

---

**Require:** observed data $\mathbf{X}$, initial parameters $\theta_e, \theta_d$ and $\theta_v$, two empty buffers $\mathcal{D}$ and $\mathcal{D}_{score}$, initial value (negative infinite) BestScore and a random ordering BestOrdeing.
  1: **while** not terminated **do**
  2:   draw a batch of samples from $\mathbf{X}$, encode them to $\mathcal{S}$ and calculate the initial state $\hat{s}_0$
  3:   **for** $t = 0, 1, \ldots, T$ **do**
  4:     collect a batch of data $\langle \hat{s}_t, a_t, r_t \rangle$ with $\pi_\theta$: $\mathcal{D} = \mathcal{D} \cup \{\langle \hat{s}_t, a_t, r_t \rangle\}$
  5:     **if** $\langle v_t, \Pi_{\prec v_t}, r_t \rangle$ is not in $\mathcal{D}_{score}$ **then**
  6:       store $\langle v_t, \Pi_{\prec v_t}, r_t \rangle$ in $\mathcal{D}_{score}$ to avoid repeated computations
  7:     **end if**
  8:   **end for**
  9:   update $\theta_e, \theta_d, \theta_v$ according to Section 4.2
 10:   **if** $\sum_{t=0}^{T} r_t >$ BestScore **then**
 11:     update the BestScore and BestOrdering
 12:   **end if**
 13: **end while**
 14: get the final DAG by pruning the BestOrdering

---

can only apply approximate model functions and also need to assume certain distribution family for caculating the reward.

Our method is summarized in Algorithm 1. In addition, we record the decomposed scores for each variable $v_j$ with different parental sets $\Pi_{\prec v_j}$ to avoid repeated computations which are generally time-consuming (see Section 4.4). Although we cannot guarantee to find the optimal ordering because policy gradient can at most guarantee local convergence (Sutton et al., 2000) and also we only have access to the empirical log-likelihood, we remark that the ordering obtained from CORL still enjoy a good performance in the experiments, compared with consistent methods like GES and PC.

### 4.3 Variable Selection

If an estimated ordering $\hat{\Pi}$ is consistent, then we obtain a fully-connected DAG (super-DAG) $\mathcal{G}^{\hat{\Pi}}$ of the underlying DAG $\mathcal{G}$. One can then pursue consistent estimation of intervention distributions based on $\hat{\Pi}$ without any additional need to find the true underlying DAG $\mathcal{G}^*$ (Bühlmann et al., 2014). For other purposes, however, we need to recover the true graph from the fully-connected DAG. There exist several efficient methods such as sparse candidate (Teyssier & Koller, 2005), significance testing of covariates (Bühlmann et al., 2014), the group Lasso (Ravikumar et al., 2009), or its improved version with a sparsity-smoothness penalty proposed in Meier et al. (2009).

For linear data models, we apply linear regression to the obtained fully-connected DAG, followed by thresholding to prune edges with small weights, as similarly used by Zheng et al. (2018); Yu et al. (2019); Zhu et al. (2020). For the nonlinear model, we follow the pruning process used by Bühlmann et al. (2014); Lachapelle et al. (2020). Specifically, for each variable $X_j$, one can fit a generalized additive model against the current parents of $X_j$ and then apply significance testing of covariates, declaring significance if the reported p-values are lower or equal to $0.001$.

### 4.4 Computational Complexity

To learn an ordering, CORL relies on the proper training of RL model. Policy gradient and stochastic gradient are adopted to train the actor and critic respectively, which are the standard choice in RL (Konda & Tsitsiklis, 2000). Similar to RL-BIC2 (Zhu et al., 2020), CORL requires the evaluation of the rewards at each episode with $\mathcal{O}(dm^2 + d^3)$ computational cost if linear functions are adopted to model the causal relations, but does not need to compute the matrix exponential term with $\mathcal{O}(d^3)$ cost due to the use of ordering search. In addition, CORL formulates causal discovery as a multi-stage decision process and we observe that CORL performs fewer episodes than RL-BIC2 before the episode reward converges (see Appendix C). We suspect that due to a significant reduction in the size of action space, the model complexity of the RL policy is reduced, thus leading to higher sample efficiency. The evaluation of Transformer encoder and LSTM decoder in CORL take $\mathcal{O}(nd^2)$ and

$\mathcal{O}(dn^2)$, respectively. However, we find that computing rewards is more dominating in the total runing time (e.g., around $95\%$ and $87\%$ for 30- and 100-node linear data models, respectively). Speeding up the calculation of rewards would be helpful in extend our approach to a larger problem, which is left as a future work.

In contrast with typical RL applications, we treat RL here as a search strategy for causal discovery, aiming to find an ordering that achieves the best score and then applying a variable selection method to remove redundant edges. Nevertheless, for the pretraining part with the goal of a good initialization, we may want sufficient generalization ability and hence consider diverse datasets with different number of nodes, noise types, causal relationships, etc.

## 5 EXPERIMENTS

In this section, we evaluate our methods against a number of methods on synthetic datasets with linear and non-linear causal relationships and also a real dataset. Specifically, these baselines include ICA-LiNGAM (Shimizu et al., 2006), three ordering-based approaches L1OBS (Schmidt et al., 2007), CAM (Bühlmann et al., 2014) and A* Lasso (Xiang & Kim, 2013), some recent gradient-based approaches NOTEARS (Zheng et al., 2018), DAG-GNN (Yu et al., 2019) and GraN-DAG (Lachapelle et al., 2020), and the RL-based approach RL-BIC2 (Zhu et al., 2020). For all the compared algorithms, we use their original implementations (see Appendix B.1 for details) and pick the recommended hyper-parameters unless otherwise stated.

Different types of data are generated in synthetic datasets which vary along five dimensions: level of edge sparsity, graph type, number of nodes, causal functions and sample size. Two types of graph sampling schemes: Erdös–Rényi (ER) and Scale-free (SF) are considered in our experiments. We denote $d$-node ER or SF graphs with on average $hd$ edges as ER$h$ or SF$h$. Two common metrics are considered: True Positive Rate (TPR) and Structural Hamming Distance (SHD). The former indicates the probability of correctly finding the positive edges among the discoveries (Jain et al., 2017). Hence, it can be used to measure the quality of an ordering, the higher the better. The latter counts the total number of missing, falsely detected or reversed edges, the smaller the better.

### 5.1 LINEAR DATA MODELS WITH GAUSSIAN AND NON-GAUSSIAN NOISE

We evaluate the proposed methods on Linear Gaussian (LG) with equal variance Gaussian noise and LiNGAM data models, and the true DAGs in both cases are known to be identifiable (Peters & Bühlmann, 2014; Shimizu et al., 2006). We set $h \in \{2, 5\}$ and $d \in \{30, 50, 100\}$ to generate observed data following the procedure done in Zheng et al. (2018) (see Appendix B.2 for details). For variable selection, we set the thresholding as 0.3 and apply it to the estimated coefficients.

Table 1: Empirical results for ER and SF graphs of 30 nodes with LG data.

| | | RANDOM | NOTEARS | DAG-GNN | RL-BIC2 | L1OBS | A* Lasso | CORL-1 | CORL-2 |
|---|---|---|---|---|---|---|---|---|---|
| ER2 | TPR | 0.41 (0.04) | 0.95 (0.03) | 0.91 (0.05) | 0.94 (0.05) | 0.78 (0.06) | 0.88 (0.04) | **0.99 (0.02)** | **0.99 (0.01)** |
| | SHD | 140.4 (36.7) | 14.2 (9.4) | 26.5 (12.4) | 17.8 (22.5) | 85.2 (23.8) | 35.3 (14.3) | **5.2 (7.4)** | **4.4 (3.5)** |
| ER5 | TPR | 0.43 (0.03) | **0.93 (0.01)** | 0.85 (0.11) | 0.91 (0.03) | 0.74 (0.04) | 0.84 (0.05) | **0.94 (0.03)** | **0.95 (0.03)** |
| | SHD | 210.2 (43.5) | **35.4 (7.3)** | 68.0 (39.8) | 45.6 (13.3) | 98.6 (32.7) | 71.2 (21.5) | **37.4 (16.9)** | 37.6 (14.5) |
| SF2 | TPR | 0.58 (0.02) | 0.98 (0.02) | 0.92 (0.09) | 0.99 (0.02) | 0.83 (0.04) | 0.93 (0.02) | **1.0 (0.01)** | **1.0 (0.01)** |
| | SHD | 118.4 (12.3) | 6.1 (2.3) | 36.8 (33.1) | 3.2 (1.7) | 49.7 (28.1) | 27.3 (18.4) | **0.0 (0.0)** | **0.0 (0.0)** |
| SF5 | TPR | 0.44 (0.03) | 0.94 (0.03) | 0.89 (0.09) | 0.96 (0.03) | 0.79 (0.04) | 0.88 (0.03) | **1.00 (0.00)** | **1.00 (0.00)** |
| | SHD | 165.4 (10.6) | 23.3 (6.9) | 47.8 (35.2) | 11.3 (5.2) | 89.3 (25.7) | 40.5 (19,8) | **0.0 (0.0)** | **0.0 (0.0)** |

Tables 1 &2 present results only for 30- and 100-node LG data models since the conclusions do not change with graphs of 50 nodes (see Appendix D for 50-node graphs). We report the performance of the popular ICA-LiNGAM, GraN-DAG and CAM in Appendix D since they are almost never on par with the best methods presented in this section. CORL-1 and CORL-2 achieve consistently good results on the LiNGAM datasets which are reported in Appendix E due to the space limit.

We now examine Tables 1 &2 (the values in parentheses represent the standard deviation across datasets per task). We can see that, across all settings, CORL-1 and CORL-2 are the best performing methods, both in terms of TPR and SHD, while NOTEARS and DAG-GNN are not too far behind.

Table 2: Empirical results for ER and SF graphs of 100 nodes with LG data.

|  |  | RANDOM | NOTEARS | DAG-GNN | RL-BIC2 | L1OBS | A* Lasso | CORL-1 | CORL-2 |
|---|---|---|---|---|---|---|---|---|---|
| ER2 | TPR | 0.33 (0.05) | 0.93 (0.02) | 0.93 (0.03) | 0.02 (0.01) | 0.54 (0.02) | 0.86 (0.04) | **0.98 (0.02)** | **0.98 (0.01)** |
|  | SHD | 491.4 (17.6) | 72.6 (23.5) | 66.2 (19.2) | 270.8 (13.5) | 481.2 (49.9) | 128.5 (38.4) | **24.8 (10.1)** | **18.6 (5.7)** |
| ER5 | TPR | 0.34 (0.04) | **0.91 (0.01)** | 0.86 (0.16) | 0.08 (0.03) | 0.53 (0.02) | 0.82 (0.05) | **0.93 (0.02)** | **0.94 (0.03)** |
|  | SHD | 984.4 (35.7) | **170.3 (34.2)** | 236.4 (36.8) | 421.2 (46.2) | 547.9 (63.4) | 244.0 (42.3) | 175.3 (18.9) | **164.8 (17.1)** |
| SF2 | TPR | 0.48 (0.03) | 0.98 (0.01) | 0.89 (0.14) | 0.04 (0.02) | 0.57 (0.03) | 0.92 (0.03) | **1.00 (0.00)** | **1.00 (0.00)** |
|  | SHD | 503.4 (23.8) | 2.3 (1.3) | 156.8 (21.2) | 281.2 (17.4) | 377.3 (53.4) | 54.0 (22.3) | **0.0 (0.0)** | **0.0 (0.0)** |
| SF5 | TPR | 0.47 (0.04) | 0.95 (0.01) | 0.87 (0.15) | 0.05 (0.03) | 0.55 (0.04) | 0.89 (0.03) | **0.97 (0.02)** | **0.98 (0.01)** |
|  | SHD | 891.3 (19.4) | 90.2 (34.5) | 165.2 (22.0) | 405.2 (77.4) | 503.7 (56.4) | 114.0 (36.4) | **19.4 (5.2)** | **10.8 (6.1)** |

As we discussed previously, RL-BIC2 only achieves satisfactory results on graphs with 30 nodes. The TPR of L1OBS is lower than that of A* Lasso, which indicates that L1OBS using greedy hill-climbing with tabu lists may not find a good ordering. Note that SHD of L1OBS and A* Lasso reported here are the results of applying our pruning method. We observe that SHD of them is greatly improved by pruning. For example, specifically, SHD of L1OBS decreases from 171.6 (29.5), 588.0 (66.2) and 1964.5 (136.6) to 85.2 (23.8), 215.4 (26.3) and 481.2 (49.9) in LG ER2 with 30, 50 and 100 nodes respectively, while TPR almost did not degraded. It shows the effectiveness of pruning by model fitting with a thresholding.

We further evaluate our method on 150-node LG ER2 data models. On the datasets, CORL-1 has TPR and SHD being 0.95 (0.01) and 63.7 (9.1), while CORL-2 has 0.97 (0.01) and 38.3 (14.3), respectively. CORL-2 outperforms NOTEARS with 0.94 (0.02) and 50.8 (21.8).

Although both CORL-1 and CORL-2 achieve the desired performance, one can notice that CORL-2 achieves slightly better SHD than CORL-1. This is a little different from the usual understanding that RL is usually easier to learn from dense rewards than from episodic reward case. We take the 100-node LG ER2 data models as an example to show the training reward curves of the CORL-1 and CORL-2 in Figure 3. One can notice that CORL-2 converges faster to a better result than CORL-1, which corresponds to the fact that CORL-2 achieves the slightly better TPR and SHD than CORL-1. We conjecture that this is because it is difficult for the critic to learn to predict the score accurately for each state; in the episodic reward case, however, it is only required to learn to predict the score accurately for the initial state.

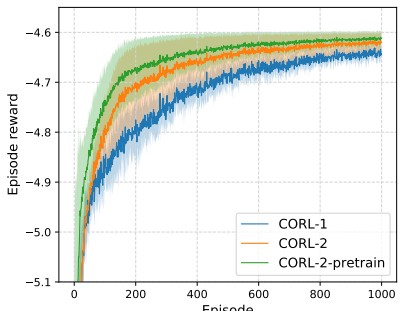

Figure 3: Learning curves of CORL-1, CORL-2 and CORL-2-pretrain on 100-node linear Gaussian dataset.

**Pretaining** We show the training reward curve of CORL-2-pretrain in Figure 3, which is CORL-2 based on a pretrained model trained in the supervised way. The testing data task is unseen in the datasets for pretraining which contain 30-node LiNGAM ER2, 50-node LiNGAM ER2, 30-node LiNGAM SF2 and 30-node GP ER1. Obviously compared to that of CORL-2 using random initialization (CORL-2), the use of a pretrained model can accelerate the model learning process. Consistent conclusion can be drawn from the experiment of CORL-1, see Appendix G. In addition, we consider using the policy model learned only on 30-node LiNGAM data model as the pretrained model on 100-node LG task. We observe that the performance is similar to that of using the pretrained model obtained in the supervised way, we do not report the results repeatedly here.

## 5.2 Non-Linear Model with Gaussian Process

In this experiment, we consider to use Gaussian Process (GP) to model causal relationships in which each causal relation $f_j$ is a function sampled from a GP with radial basis function kernel of bandwidth one and normal Gaussian noises, which is known to be identifiable according to Peters et al. (2014). We set $h = 1$ and $4$ to get ER1 and ER4 graphs, respectively, and generate data by exploiting them (see Appendix B.2 for details). Our method is evaluated on different sample numbers, the results when $m = 500$ are reported here (see Appendix F for additional results). Note that due to the efficiency of

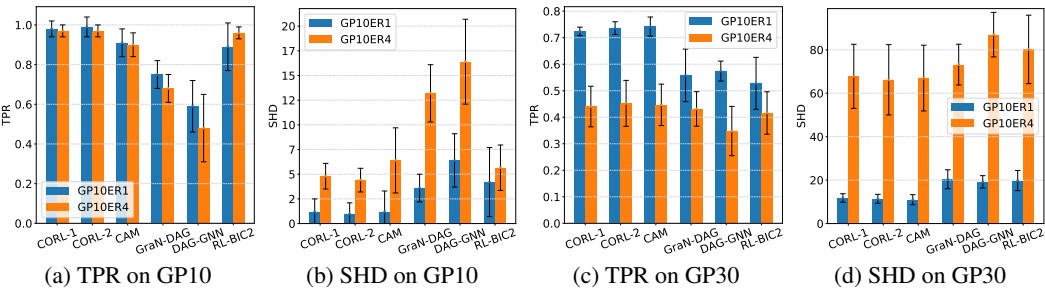

| (a) TPR on GP10 | (b) SHD on GP10 | (c) TPR on GP30 | (d) SHD on GP30 |

Figure 4: The empirical results on GP data models with 10 and 30 nodes.

the reward calculation, we only experimented on nonlinear data models of up to 30 nodes scale. The pruning method for variable selection used here is the CAM pruning from Bühlmann et al. (2014).

The results on 10-node and 30-node datasets with ER1 and ER4 graphs are shown in Figure 3. Here we only consider some baselines that are competitive in the nonlinear data models, where CAM is a very strong ordering based baseline. Although GraN-DAG achieves better results than DAG-GNN, they are worse than CAM overall. We believe this is because $500$ samples are so small that GraN-DAG and DAG-GNN have not learned the good model. RL-BIC2 performs well on 10-node datasets, but achieves poor results on 30-node datasets, probably due to its lack of scalability. CAM, CORL-1 and CORL-2 have good results, with CORL-2 achieving the best results on 10-node graphs and slightly worse than CAM on 30-node graphs. All of these methods have better results on ER1 than on ER4, especially on 30-node graphs.

## 5.3 REAL DATA

The Sachs dataset (Sachs et al., 2005), with $11$-node and $17$-edge true graph, is widely used for research on graphical models. The expression levels of protein and phospholipid in the dataset can be used to discover the implicit protein signal network. The observational dataset has $m = 853$ samples and is used to discover the causal structure. GP is used to model the causal relationship in our method. In this experiment, CORL-1, CORL-2 and RL-BIC2 achieve the best SHD $11$. CAM, GraN-DAG, and ICA-LiNGAM achieve the SHD $12$, $13$ and $14$, respectively. Particularly, DAG-GNN and NOTEARS result in SHD $16$ and $19$, respectively, whereas an empty graph has an SHD $17$.

## 6 CONCLUSION

In this work, we have proposed a RL-based approach for causal discovery named CORL. It searches the space of variable orderings, instead of the space of directed graphs. We formulate ordering search as an MDP and have further proposed CORL-1 and CORL-2 for training the ordering generating model. For a generated ordering, it can be pruned by variable selection to the final DAG. The empirical results on the synthetic and the real datasets show that our approach is promising.

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

APPENDIX

# A NETWORK ARCHITECTURES AND HYPER-PARAMETERS

## A.1 MULTIPLE NETWORK ARCHITECTURE DESIGNS FOR ENCODER AND DECODER

There are a variety of neural network modules for encoder and decoder structures. Here we consider some representative modules: including , Multi-layer Perceptrons (MLP) module, an LSTM based recurrent neural network module, and the self-attention based encoder in the Transformer. In addition, we use the original observation data as the state directly, i.e., no encoder module is used, to show the necessity of encoder, which is denoted as Null. MLP consists of 3-layer feed-forward neural networks with 256, 512 and 256 units. The architecture of LSTM and Transformer are that introduced in Appendix A.2.

The empirical results of CORL-2 on 30-node LG ER2 datasets are reported in Table 3. We observe that LSTM decoder achieves a better performance than that of MLP decoder. This shows that LSTM is more effective than MLP in sequential decision tasks. Besides, the overall performance of neural network encoder is better than that of Null, which shows that the data pre-processed by encoder module is necessary. Among all these encoders, Transformer encoder achieves the best results. Similar conclusion was drawned in Zhu et al. (2020). We hypothesize that the performance of Transformer encoder is benefit from the self-attention scheme.

Table 3: Empirical results of CORL-2 with different choices of encoder and decoder on 30-node LG ER2 datasets. The smaller SHD the better, the higher TPR the better.

|  |  | Encoder | | | |
|---|---|---|---|---|---|
|  |  | Null | LSTM | MLP | Transformer |
| MLP Decoder | TPR | 0.81 (0.07) | 0.86 (0.10) | 0.96 (0.02) | 0.98 (0.02) |
|  | SHD | 54.2 (25.1) | 36.0 (26.7) | 11.0 (5.3) | 5.0 (3.3) |
| LSTM Decoder | TPR | 0.94 (0.04) | 0.88 (0.09) | 0.97 (0.01) | 0.99 (0.01) |
|  | SHD | 20.6 (20.0) | 29.0 (17.8) | 8.6 (4.3) | 4.4 (3.5) |

## A.2 NETWORK ARCHITECTURE AND HYPER-PARAMETERS OF CORL

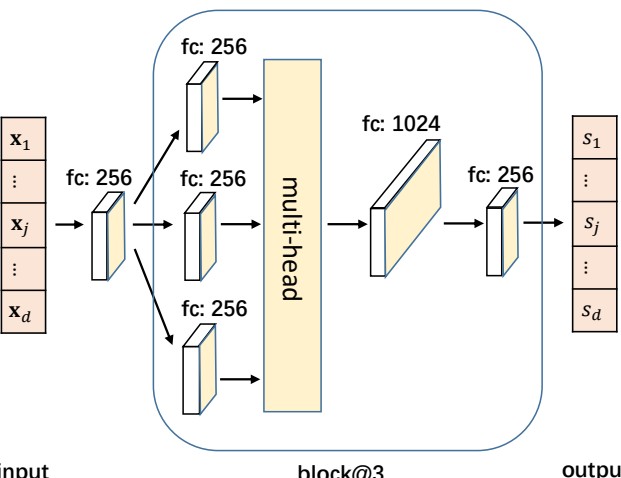

Figure 5: Illustration of the Transformer encoder. The encoder embeds the observed data $x_j$ of each variable $j$ into the state $s_j$. Notation block@3 denotes three blocks here.

Both CORL-1 and CORL-2 use the actor-critic algorithm. We use the Adam optimizer with learning rate $1e-4$ and $1e-3$ for actor and critic respectively. The discount factor $\gamma$ is set to $0.98$. The actor

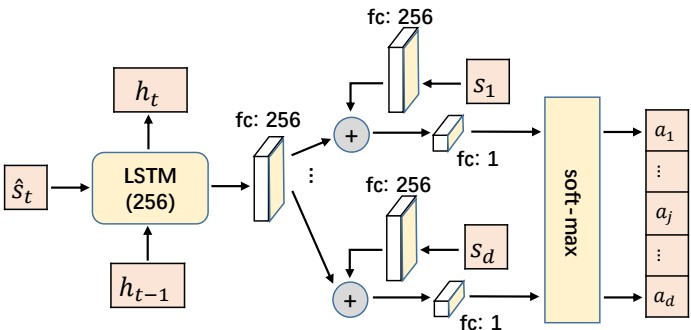

Figure 6: Illustration of the LSTM decoder. At each time step, it maps the state $\hat{s}_t$ to a distribution over action space $\mathcal{A} := \{a_1, \ldots, a_d\}$, then an action (variable) can be selected randomly according to the distribution.

consists of an encoder and a decoder. We illustrate the neural network structure of the Transformer encoder used in our experiments in Figure 5. It consists of a feed-forward layer with 256 units and three blocks. Each block is composed of a multi-head attention network with 8 heads and 2-layer feed-forward neural networks with 1024 and 256 units. Each feed-forward layer is followed by a normalization layer. Given a batch of observed samples with shape $b \times d \times n$, where $b$ denotes the batch size, $d$ the node number and $n$ the number of observed data for each variable in a batch, the final output of the encoder is a batch of embedded state with shape $b \times d \times 256$.

We illustrate the neural network structure of the decoder in Figure 6, which is mainly a LSTM similar to the decoder proposed by Vinyals et al. (2015). The LSTM takes a state as input and outputs a embedding. The embedding is mapped to the action space $\mathcal{A}$ by using some feed-forward neural networks, a soft-max module and the pointer mechanism (Vinyals et al., 2015). The outputs of the encoder are processed as the initial hidden state $h_0$ of the decoder. The LSTM with 256 hidden units is used here. All of the feed-forward neural networks used in decoder have 256 units.

The critic uses 3-layer feed-forward neural networks with 512, 256 and 1 units. It takes a state $\hat{s}$ as input and outputs a predicted value for the current policy given state $\hat{s}$. For CORL-1, the critic needs to predict the score for each state. For CORL-2, the critic takes the initial state $\hat{s}_0$ as input and outputs a predicted value directly for a complete ordering.

## B  BASELINES AND DATE SETS

### B.1  DETAILS OF BASELINES

The details of all the baselines considered in our experiments are listed as follows:

- ICA-LiNGAM assumes linear non-Gaussian additive model for data generating procedure and applies independent component analysis to recover the weighted adjacency matrix. This method can usually achieve good performance on LiNGAM datasets. However, it does not provide guarantee for linear Gaussian datasets. [1]

- NOTEARS recovers the causal graph by estimating the weighted adjacency matrix with the least squares loss and the smooth characterization for acyclicity constraint. [2]

- DAG-GNN formulates causal discovery in the framework of variational autoencoder and optimizes a weighted adjacency matrix with the evidence lower bound and a modified smooth characterization on acyclicity as loss function. [3]

---

[1] https://sites.google.com/site/sshimizu06/lingam
[2] https://github.com/xunzheng/notears
[3] https://github.com/fishmoon1234/DAG-GNN

- GraN-DAG models the conditional distribution of each variable given its parents with feed-forward neural networks. It uses the smooth directed acyclic constraint from NOTEARS to find a DAG that maximizes the log-likelihood of the observed samples. [4]

- RL-BIC2 formulates the causal discovery as a one-step decision making process and uses score function and acyclic constraint from NOTEARS to calculate the reward for the recovered directed graph. [5]

- CAM conducts a greedy estimation procedure that starts with an empty DAG and adds at each iteration the edge $(v_k, v_j)$ between nodes $v_k$ and $v_j$ that corresponds to the largest gain in log-likelihood. For a searched ordering, they prune it to the final DAG by applying significance testing of covariates. They perform the preliminary neighborhood selection to reduce the ordering space size searched. [6]

- L1OBS performs heuristic search (greedy hill-climbing with tabu lists) through the space of topological orderings to search a ordering with the best score, then it uses L1 variable selection to pruning the searched ordering (fully-connected DAG) to the final DAG. [7]

- A* Lasso with a limited queue size incorporates a heuristic scheme into a dynamic programming based method. It first prunes the search space by using A* Lasso, then it further pruning the search space by limiting the size of the priority queue in the OPEN list of A* Lasso. The queue size usually needs to be adjusted to balance the time cost and the quality of the solution. [8]

## B.2 DETAILS ON DATASETS GENERATION

We generate synthetic datasets which vary along five dimensions: level of edge sparsity, graph type, number of nodes, causal functions and sample size. We sampled 5 datasets of the required number examples for each task as follows: a ground truth DAG $\mathcal{G}$ is drawn randomly from either the Erdös–Rényi (ER) or Scale-free (SF) graph model; then, the data is generated according to a specific sampling scheme.

Specifically, for Linear Gaussian (LG) case, we set $h \in \{2, 5\}$ and $d \in \{30, 50, 100\}$ to obtain the ER graph and SF graph (different types) with different levels of edge sparsity and different number of nodes, respectively. Then 5 datasets of 3000 examples are generated for each task following $\mathbf{X} = W^T \mathbf{X} + \epsilon$, where $W \in \mathbb{R}^{d \times d}$ denotes the weight matrix which is obtained by assigning edge weights independently from Unif($[-2, -0.5] \cup [0.5, 2]$). Note that $\epsilon$'s are standard Gaussian noises with equal variances for each variable so as to LG data model is identifiable (Peters & Bühlmann, 2014).

For LiNGAM data model, the datasets are generated in the similar way with LG but the sampling for $\epsilon$ are different. The non-Gaussian noises are obtained by following Shimizu et al. (2006) which passes the noise samples from Gaussian distribution through a power nonlinearity to make them non-Gaussian. LiNGAM is identifiable shown in Shimizu et al. (2006).

Another data generating process is GP. We first obtain graph with different density and different number of nodes. Then the datasets with different sample sizes are generated following $X_j = f_j(\text{Pa}(X_j)) + \epsilon_j$ with jointly independent Gaussian noises for all $j$ where the function $f_j$ is a function sampled from a GP with radial basis function kernel of bandwidth one and normal Gaussian noises. This setting is known to be identifiable according to Peters et al. (2014). Note that due to the efficiency of the reward calculation, we only experimented on nonlinear data models of up to 30 nodes.

## C   TOTAL NUMBER OF EPISODES BEFORE CONVERGENCE

Table 4 reports the total number of episodes required for CORL-2 and RL-BIC2 to be converged, averaged over five datasets. Note that the episodic reward is evaluated once per episode. CORL

---

[4]`https://github.com/kurowasan/GraN-DAG`
[5]`https://github.com/huawei-noah/trustworthyAI/tree/master/Causal_`
`Structure_Learning/Causal_Discovery_RL`
[6]`https://cran.r-project.org/web/packages/CAM.`
[7]`https://www.cs.ubc.ca/~murphyk/Software/DAGlearn/`
[8]`http://www.cs.cmu.edu/~jingx/software/AstarLasso.zip`

formulates causal discovery as a multi-stage decision process and we observe that CORL performs fewer episodes than RL-BIC2 before the episode reward converges. We suspect that due to a significant reduction in the size of action space, the model complexity of the RL policy is reduced thus leading to higher sample efficiency. Some runtimes about them are also provided here (CORL-2 total runtime $\approx$ 15 minutes against RL-BIC2 $\approx$ 3 hours for 30-node ER2 graphs, $\approx$ 4 hours against $\approx$ 14 hour for 50-node ER2 graphs, and CORL-2 $\approx$ 7 hours for 100-node ER2 graphs). We set the maximal runtime up to 24 hours, but RL-BIC2 did not converge within that time on 100-node graphs, hence we did not report it here. Note that these runtime may be significantly reduced by parallelizing the evaluation of reward.

Table 4: Total number of iterations ($\times 10^3$) before RL converge on LG data.

|  | 30 nodes | | 50 nodes | | 100 nodes | |
| --- | --- | --- | --- | --- | --- | --- |
|  | ER2 | ER5 | ER2 | ER5 | ER2 | ER5 |
| CORL-2 | 1.0 (0.3) | 1.1 (0.4) | 1.9 (0.3) | 2.4 (0.3) | 2.3 (0.5) | 2.9 (0.4) |
| RL-BIC2 | 3.9 (0.5) | 4.1 (0.6) | 3.4 (0.4) | 3.5 (0.5) | $\times$ | $\times$ |

## D  ADDITIONAL RESULTS ON LINEAR GAUSSIAN DATASETS

The results for 50-node LG data models are presented in Table 5. The conclusion is similar to the 30- and 100-node experiments. The results of ICA-LiNGAM, GraN-DAG and CAM on LG data models are presented in Table 6. Their performances do not compare favorably to CORL-1 nor CORL-2 in LG datasets.

It is not surprising that ICA-LiNGAM does not perform well because the algorithm is specifically designed for non-Gaussian noise and does not provide guarantee for LG data models. We hypothesize that CAM's poor performance on LG data models is because it uses nonlinear regression instead of linear regression. As for GraN-DAG, it uses 2-layer feed-forward neural networks to model the causal relationships, which may not be able to learn a good linear relationship in this experiment.

Table 5: Empirical results for ER and SF graphs of 50 nodes with LG data. The higher TPR the better, the smaller SHD the better.

|  |  | RANDOM | NOTEARS | DAG-GNN | RL-BIC2 | L1OBS | A* Lasso | CORL-1 | CORL-2 |
| --- | --- | --- | --- | --- | --- | --- | --- | --- | --- |
| ER2 | TPR | 0.31 (0.03) | 0.94 (0.02) | 0.94 (0.04) | 0.79 (0.10) | 0.56 (0.02) | 0.88 (0.03) | **0.97 (0.04)** | **0.97 (0.02)** |
|  | SHD | 295.4 (28.5) | 38.6 (10.8) | 30.6 (8.3) | 88.5 (49.3) | 288.0 (66.2) | 154.3 (27.6) | 24.0 (32.3 | **17.9 (10.6)** |
| ER5 | TPR | 0.32 (0.02) | **0.90 (0.01)** | 0.87 (0.14) | 0.74 (0.03) | 0.57 (0.03) | 0.82 (0.03) | 0.90 (0.02) | **0.92 (0.02)** |
|  | SHD | 378.4 (24.2) | **67.8 (7.5)** | 93.2 (109.4) | 128.9 (40.4) | 299.4 (53.6) | 104.0 (28.3) | 68.3 (10.2) | **64.8 (13.1)** |
| SF2 | TPR | 0.49 (0.04) | 0.99 (0.01) | 0.90 (0.13) | 0.84 (0.05) | 0.67 (0.02) | 0.89 (0.03) | **1.00 (0.00)** | **1.00 (0.00)** |
|  | SHD | 215.6 (14.7) | 3.5 (1.6) | 79.3 (93.2) | 115.2 (57.4) | 182.3 (33.4) | 124.0 (35.2) | **0.0 (0.0)** | **0.0 (0.0)** |
| SF5 | TPR | 0.51 (0.03) | **0.94 (0.12)** | 0.88 (0.12) | 0.75 (0.05) | 0.61 (0.03) | 0.81 (0.02) | 0.94 (0.03) | **0.95 (0.02)** |
|  | SHD | 345.6 (24.3) | **20.1 (14.3)** | 89.2 (99.2) | 115.2 (57.4) | 217.3 (36.4) | 131.0 (25.3) | 24.3 (11.2) | **20.8 (10.1)** |

## E  RESULTS ON 30-, 50- AND 100-NODE LiNGAM DATASETS

Here, we report the empirical results on 30-, 50- and 100-node LiNGAM datasets in Table 7. The observed samples are generated according to the same procedure with linear Gaussian datasets (see Appendix B.2 for details). The non-Gaussian noise is obtained by passing the noise samples from Gaussian distribution through a power nonlinearity to make them non-Gaussian. For L1OBS, we increased the authors' recommended number of evaluations 2500 to 10000. For A* Lasso, we set the queue size to 10, 500 and 1000, and report the best result for all of these different parameter settings. The results of L1OBS and A* Lasso reported here are that of pruning using our pruning method. For other baselines, we pick the recommended hyper-parameters.

Among all these algorithms, ICA-LiNGAM can recover the true graph on most of the LiNGAM data models. This is because ICA-LiNGAM is specifically designed for non-Gaussian noise. CORL-1 and CORL-2 achieve consistently good results than other baselines.

Table 6: Empirical results of ICA-LiNGAM, GraN-DAG and CAM (against CORL-2 for reference) for ER and SF graphs with LG data. The higher TPR the better, the smaller SHD the better.

|  |  |  | ICA-LiNGAM | GraN-DAG | CAM | CORL-2 |
|---|---|---|---|---|---|---|
| 30 nodes | ER2 | TPR | 0.75 (0.03) | 0.51 (0.17) | 0.47 (0.05) | **0.99 (0.01)** |
|  |  | SHD | 112.3 (12.8) | 96.0 (11.3) | 110.8 (10.3) | **4.4 (3.5))** |
|  | ER5 | TPR | 0.57 (0.03) | 0.52 (0.03) | 0.46 (0.02) | **0.95 (0.03)** |
|  |  | SHD | 161.8 (9.2) | 175.2 (27.4) | 191.3 (32.5) | **37.6 (14.5)** |
|  | SF2 | TPR | 0.58 (0.05) | 0.61 (0.04) | 0.63 (0.02) | **1.0 (0.0)** |
|  |  | SHD | 149.0 (19,8) | 136.4 (21.2) | 115.2 (26.7) | **0.0 (0.0)** |
|  | SF5 | TPR | 0.56 (0.04) | 0.58 (0.02) | 0.60 (0.03) | **1.0 (0.0)** |
|  |  | SHD | 160.5 (8.9) | 142.4 (24.3) | 122.2 (17.4) | **0.0 (0.0)** |
| 50 nodes | ER2 | TPR | 0.73 (0.03) | 0.11 (0.04) | 0.55 (0.06) | **0.97 (0.02)** |
|  |  | SHD | 108.8 (11.3) | 173.0 (22.9) | 140.8 (35.4) | **17.9 (10.6)** |
|  | ER5 | TPR | 0.57 (0.01) | 0.64 (0.03) | 0.61 (0.02) | **0.92 (0.02)** |
|  |  | SHD | 199.8 (90.7) | 154.2 (36.4) | 178.3 (34.8) | **64.8 (13.1)** |
|  | SF2 | TPR | 0.59 (0.04) | 0.44 (0.05) | 0.57 (0.02) | **1.00 (0.00)** |
|  |  | SHD | 208.5 (83.2) | 158.6 (34.5) | 131.2 (24.4) | **0.0 (0.0)** |
|  | SF5 | TPR | 0.57 (0.01) | 0.49 (0.04) | 0.53 (0.03) | **0.95 (0.02)** |
|  |  | SHD | 216.6 (88.4) | 243.9 (27.2) | 235.2 (34.2) | **20.8 (10.1)** |
| 100 nodes | ER2 | TPR | 0.73 (0.02) | 0.38 (0.02) | 0.43 (0.02) | **0.98 (0.01)** |
|  |  | SHD | 268.4 (28.5) | 191.3 (31.9) | 126.4 (27.8) | **18.6 (5.7)** |
|  | ER5 | TPR | 0.57 (0.05) | 0.42 (0.03) | 0.47 (0.02) | **0.94 (0.03)** |
|  |  | SHD | 311.1 (63.7) | 208.2 (54.4) | 182.3 (34.9) | **164.8 (17.1)** |
|  | SF2 | TPR | 0.69 (0.03) | 0.40 (0.03) | 0.44 (0.02) | **1.00 (0.00)** |
|  |  | SHD | 367.6 (67.5) | 239.9 (43.2) | 35.2 (37.4) | **0.0 (0.0)** |
|  | SF5 | TPR | 0.57 (0.05) | 0.39 (0.03) | 0.48 (0.04) | **0.98 (0.01)** |
|  |  | SHD | 362.3 (82.8) | 219.3 (32.2) | 125.2 (24.7) | **10.8 (6.1)** |

Table 7: Empirical results on 30-, 50- and 100-node LiNGAM ER2 datasets. The smaller SHD the better, the higher TPR the better.

| | 30 nodes ER2 | | 50 nodes ER2 | | 100 nodes ER2 | |
|---|---|---|---|---|---|---|
| Method | TPR | SHD | TPR | SHD | TPR | SHD |
| **ICA-LiNGAM** | **1.00 (0.00)** | **0.0 (0.0)** | **1.00 (0.00)** | **0.0 (0.0)** | **1.00 (0.00)** | **1.0 (0.9)** |
| NOTEARS | 0.94 (0.04) | 17.2 (13.2) | 0.95 (0.02) | 33.2 (16.5) | 0.94 (0.03) | 69.2 (23.2) |
| DAG-GNN | 0.94 (0.03) | 19.6 (10.5) | 0.96 (0.01) | 24.6 (2.9) | 0.93 (0.03) | 66.2 (19.2) |
| GraN-DAG | 0.28 (0.09) | 100.8 (14.6) | 0.20 (0.01) | 177.0 (25.9) | 0.16 (0.04) | 312.8 (25.2) |
| RL-BIC2 | 0.94 (0.07) | 19.8 (23.0) | 0.80 (0.12) | 86.0 (51.9) | 0.13 (0.12) | 291.3 (24.1) |
| CAM | 0.60 (0.11) | 310.0 (34.0) | 0.33 (0.07) | 178.0 (31.9) | 0.53 (0.05) | 247.2 (32.1) |
| L1OBS | 0.72 (0.04) | 85.3 (23.3) | 0.47 (0.02) | 212.6 (24.6) | 0.41 (0.03) | 470.5 (48.1) |
| A* Lasso | 0.87 (0.03) | 42.3 (16.3) | 0.88 (0.03) | 82.6 (17.6) | 0.85 (0.04) | 102.5 (22.6) |
| **CORL-1** | **0.99 (0.01)** | **3.8 (6.4)** | **0.96 (0.06)** | **24.6 (37.7)** | **0.98 (0.01)** | **20.0 (7.9)** |
| **CORL-2** | **0.99 (0.01)** | **3.9 (5.6)** | **0.96 (0.08)** | **20.2 (11.3)** | **0.99 (0.01)** | **13.8 (7.2)** |

## F  RESULTS ON 20-NODE GP DATASETS WITH DIFFERENT SAMPLE SIZES

We take the 20-node GP data models as an example to show the performance of our method on different sample numbers. We set $h = 4$ to get ER4 graphs and generate data by using them. We illustrate the empirical results in Table 8. Since previous experiments have shown that CORL-2 is slightly better than CORL-1, we only report the results of CORL-2 here. CAM as the most competitive baseline, we also report its results on these datasets. TPR reported here is calculated based on the variable ordering. We can see that, as the sample size decreases, CORL-2 ends up outperforming CAM. We believe this is because CORL-2 benefits from the exploratory ability of RL.

## G  CORL-1 WITH A PRETRAINED MODEL

We show the training reward curves of CORL-1 and CORL-1-pretrain which is CORL-1 based on a pretrained model in Figure 7. To obtain a pretraining model with good generalization ability, we combine various types data described in Appendix B.2 with different levels of edge sparsity, graph

Table 8: Empirical results on 20-node GP ER1 datasets with different sample sizes. The smaller SHD the better, the higher TPR the better.

| Sample size | Name | TPR | SHD |
|---|---|---|---|
| 1000 | CAM | 0.91 (0.03) | 30.0 (3.7) |
| | CORL-2 | 0.87 (0.03) | 36.5 (3.1) |
| 500 | CAM | 0.86 (0.03) | 45.0 (2.5) |
| | CORL-2 | 0.85 (0.03) | 46.3 (2.3) |
| 400 | CAM | 0.83 (0.02) | 51.0 (2.7) |
| | CORL-2 | 0.84 (0.03) | 50.5 (3.0) |
| 200 | CAM | 0.60 (0.03) | 66.3 (1.9) |
| | CORL-2 | 0.75 (0.02) | 63.1 (1.5) |

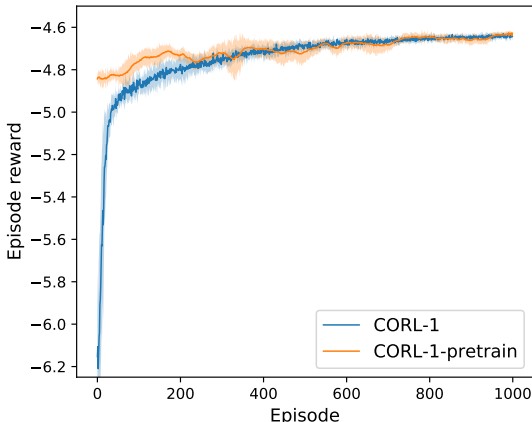

Figure 7: Learning curves of CORL-1 and CORL-1-pretrain on 100-node LG datasets.

type and number of nodes and data generating process to construct observation samples. Next, we train a policy model by supervised learning on the mixed datasets. Finally, we use the pretrained model as a start point on the task that has never been seen before during pretraining.

From Figure 7, we can observe that although the pretrained model is trained on other types of datasets, it can still accelerate the training as the initial model on the LG dataset task. This shows that our policy model may learn some implicit knowledge across different tasks.

