# OpenReview forum: "Ordering-Based Causal Discovery with Reinforcement Learning"
_ICLR.cc/2021/Conference — Reject_

### Official Review · AnonReviewer2 · 2020-10-29

**Rating:** 5
**Confidence:** 3

**Review:**

This paper describes an RL approach for DAG structure learning. In particular, the RL agent explores the space of permutations instead of space of adjacency matrices. The experimental results suggest superior performance in a few synthetic data settings.

Pros:
- Using RL for causal discovery is gaining popularity recently. But to the best of my knowledge this paper is the first to search over the space of variable orderings, instead of directly searching over the space of DAGs.
- The experimental result seems to show a significant performance improvement compared to other baselines.

Cons:
- The superior performance requires some explanation. For instance, there are many design choices made for the model, but it is hard to see which part of the model has brought the performance gain. It would be nice to see some ablation study.
- Compared to the baselines, this paper proposes a substantially more complicated model. It is surprising that it does not overfit. But it would be nice to see some justifications (other than better TPR, SHD).


Detailed comments:
This is a well written paper, but I would need more information to better evaluate it.
- The RL formulation requires more explanation. For instance, why does the state s_j really represent? Why initialize s_0 with average states? Why does the state encoder use attention networks? What is the role of state transition, if all states depend on all X?
- More diverse set of experiments would be useful: different density, different topological properties, different noise variances, and different nonlinearities, etc.
- Since the architecture is substantially more complex than baselines, it would be nice to include computational costs like runtime.

---

> ### Author Response · Authors · 2020-11-21
> **Response to Reviewer 2**
>
> We thank the reviewer for valuable comments/suggestions.
>
> **1. The superior performance requires some explanation. For instance, there are many design choices made for the model, but it is hard to see which part of the model has brought the performance gain. It would be nice to see some ablation study.**
>
> Thanks for this helpful suggestion. We have conducted additional experiments on the different choices of encoder and decoder module, which are reported in Appendix A.
> From the results, we observe that different encoder and decoder modules have different effects on the performance, their choice is the key to our model performance.  Specifically, LSTM decoder achieves the  better performance than MLP decoder, which shows that LSTM is more effective in the sequential decision task.  Among all these encoders, Transformer encoder achieves the best results. We hypothesize that the performance of Transformer encoder is benefit from  the self-attention scheme.
>
> **2. Regarding the remark: '... It is surprising that it does not overfit. But it would be nice to see some justifications (other than better TPR, SHD).'**
>
> This is insightful. In RL this is known as the exploration-exploitation trade-off. We learn a stochastic policy for generating an ordering, which outputs a probability distribution over actions. The behaviour actions are sampled from the stochastic policy, allowing RL to explore the state space without always taking the same action and handling the exploration/exploitation trade off without hard coding it.
>
> In addition, we treat the RL based approach as a *search* strategy, aiming to find an ordering that achieves the best score. After finding an ordering, we apply variable selection steps to reduce spurious edges.
> For the pretraining part with the goal of a good initialization, we may want sufficient generalization ability and hence consider diverse datasets with different number of nodes, noise types, causal relationships, etc.
>
> **3. The RL formulation requires more explanation. For instance, why does the state $s_j$ really represent? Why initialize $s_0$ with average states? Why does the state encoder use attention networks? What is the role of state transition, if all states depend on all X?**
>
> 1) State $s_j$ is an embedding of observations; the encoder learns to encode the state $s_j$ to help the decoder to generate the promising ordering based on the state. Different encoders have the different capacity to encode (represent) the state.
> 2) Although the initial state $s_0 \in \mathbb R^{n}$ can be generated randomly, we pick it as $s_0=\frac1d\sum_{i=1}^d s_i$,  considering that the source node is fixed in correct ordering. And it works well in our experiments.
> 3) We have conducted experiments to show the effect of some other encoders (Null, MLP, LSTM and Transformer). Empirical results show that the attention based encoder in the Transformer structure performs the best, possibly due to its self-attentions scheme. Please find more details in Appendix A.
> 4) In practice, we did not use all $X$, but randomly sampled minibatches for sample efficiency as described in Section 4.2. Moreover, if all $X$ is used, i.e., each state depends on the full observed data of a variable, state transition still play the previous role.  Specifically, if the selected variable is $v_j$ at the $t$-th decision step, then the state is transferred to the state $s_j \in \mathcal{S}$ which is the $j$-th output from Transformer encoder, i.e., $\hat{s}_{t+1}=s_j$.
>
> **4. More diverse set of experiments would be useful: different density, different topological properties, different noise variances, and different nonlinearities, etc.**
>
> Thanks for your suggestion. We have provided more experiments on datasets, which now vary along five dimensions: level of edge sparsity (2d and 5d edges), graph type (ER and SF), number of nodes (30, 50 and 100), causal functions (linear Gaussian, LiNGAM and Gaussian process), and sample size. Please find more details in Section 5 in the revised manuscript.
>
> **5.  Regarding : '...it would be nice to include computational costs like runtime.'**
>
> Thanks for this helpful suggestion. We have added a discussion on the computational complexity in Section 4.4 and have reported the total number of episodes to reach converge and runtime on linear Gaussian datasets in Appendix C in the revised manuscript.
> Compared to RL-BIC2, a previous RL-based method, CORL enjoys higher sample efficiency since it converges using a lower number of episodes.
> A fact is that the total time of the RL based approach is generally longer than other methods, but we believe the time is acceptable given the gained accuracy.  Furthermore, we have shown that while the adopted NN structures, like the Transformers, are complex, they also allow certain generalization capacities, so we can use pretraining to improve accelerating training.
> We are currently working to provide more about running times.

---

### Official Review · AnonReviewer4 · 2020-10-29
**Minor improvement**

**Rating:** 5
**Confidence:** 3

**Review:**

Update: I have read the authors responses and am happy that they have addressed some of my concerns in my review, especially comparisons with previous works. However, I still have the concern that the improvement seems very minor compared to previous works. Thus, I decide to keep my score unchanged.

Original review:

In this paper, the authors proposed a new RL based algorithm to discover causal DAG models from observational data. Unlike the previous RL based causal discovery algorithm that performs causal discovery via searching through DAG space, it propose to instead search through the smaller ordering space, thereby achieving better empirical performance.

Cons.

1. Ordering based causal discovery is a well-established field. In addition to the empirical findings cited in this manuscript, there are also [1,2] that theoretically understand ordering based causal discovery. New algorithms have also been proposed in the two papers. The authors should cite and compare against the two research works from both theoretical and empirical perspective, which is not the case in the current manuscript.

2. In the variable selection part, since the algorithms perform multiple hypothesis testing, the p-values should be adjusted (such as bonferroni correction) accordingly to adapt to the multiple testing setting.

3. Proposition 1 is trivial and has been proven / taken as known knowledge in many other places, such as [1].

4. The methodological improvement seems not very novel, RL based causal discovery method and ordering based method have both been intensively studied before, and combining them together seems not a very exciting improvement from my perspective.

[1] https://arxiv.org/abs/1702.03530
[2] Bernstein, Daniel, et al. "Ordering-based causal structure learning in the presence of latent variables." AISTATS 2020

---

> ### Author Response · Authors · 2020-11-21
> **Response to Reviewer 4**
>
> We thank the reviewer for the feedback on our work. We provide point by point responses below.
>
> **1. Ordering based causal discovery is a well-established field. In addition to the empirical findings cited in this manuscript, there are also [1,2] that theoretically understand ordering based causal discovery. New algorithms have also been proposed in the two papers. The authors should cite and compare against the two research works from both theoretical and empirical perspective, which is not the case in the current manuscript.**
>
> Thanks for pointing out these two works. We have added a discussion on them in Section 2, which is summarized as
>
> " sparsest permutation [3] and greedy sparest permutation [1] can guarantee consistency of Markov equivalence class, based on some conditional independence relations and certain assumptions like faithfulness. A variant of greedy sparest permutation was also proposed in [2] for the setting with latent variables. In the present work, we mainly work on identifiable cases which may have different assumptions from theirs and consider the setting without latent variable. ".
>
> Although it is beneficial to include experimental results about them, we did not find any released codes of the two algorithms so may not be able to conduct an empirical comparison.  If we omit certain releasing information, please let us know. Thanks.
>
> **2. In the variable selection part, since the algorithms perform multiple hypothesis testing, the p-values should be adjusted (such as bonferroni correction) accordingly to adapt to the multiple testing setting.**
>
> The CAM [4] based pruning method declares significance if the reported p-values are lower or equal to 0.001, which shows good empirical performance and has been adopted by several algorithms. Thus, we simply use the same method to make comparison easy and fair. We appreciate the suggestion from the reviewer and will try it in future works.
>
> **3. Proposition 1 is trivial and has been proven / taken as known knowledge in many other places, such as [1].**
>
> We agree that Proposition 1 is not hard; it is only used as a part of *Background* to demonstrate the reasoning of finding a topological ordering. Based on our understanding of [1], [1] considered faithfulness, TSP, ESP, and SMR assumptions while we assumed identifiability (of course, identifiability may require similar or identical conditions in some cases, but can also rely on different assumptions like causal minimality that is weaker than faithfulness). Also, [1] used CI relations and found consistency about Markov equivalence class. Back to Proposition 1, we however did not find an exact statement in the literature so we provide the result here with a quick proof. We will definitely modify the result if the reviewer could provided any pointers to existing works that have formally stated this result.
>
> **4. The methodological improvement seems not very novel, RL based causal discovery method and ordering based method have both been intensively studied before, and combining them together seems not a very exciting improvement from my perspective.**
>
> We would like to mention that the present work is not intended as a simple combination of two existing methods. RL has been shown promising for some causal discovery problems, but is slow and is limited to small problems. To improve the RL based approach, we came up with two directions: 1) to reduce search space and 2) to accelerate training. Our current solution to the first one is to use ordering search, but could be handled/combined by other approaches like preliminary neighbor selections. We then transfer the previous one-step episodic formulation to a multi-stage Markov decision process, and adopt pretraining to accelerate training, leading to much improvement over existing RL based approach. We hope that our work can motivate others to further improve the RL based approach wrt. efficiency and efficacy.
>
> ---
> References
>
> [1] Liam Solus, Yuhao Wang, and Caroline Uhler. Consistency guarantees for greedy permutation-based
> causal inference algorithms. arXiv preprint arXiv:1702.03530, 2017.
>
> [2] Daniel Bernstein, Basil Saeed, Chandler Squires, and Caroline Uhler. Ordering-based causal structure
> learning in the presence of latent variables. In International Conference on Artificial Intelligence
> and Statistics (AISTATS), pp. 4098–4108. PMLR, 2020.
>
> [3] Garvesh Raskutti and Caroline Uhler. Learning directed acyclic graph models based on sparsest
> permutations. Stat, 7(1):e183, 2018.
>
> [4] Peter Bühlmann, Jonas Peters, Jan Ernest, et al. CAM: Causal additive models, high-dimensional
> order search and penalized regression. The Annals of Statistics, 42(6):2526–2556, 2014.

---

### Official Review · AnonReviewer3 · 2020-10-29
**Interesting formulation, empirical evaluation raises some questions**

**Rating:** 5
**Confidence:** 4

**Review:**

The paper proposes a RL formulation to learn the DAG topological order (and then the DAG structure). The reduced search space enables a better and seemly more efficiently learning than the existing RL baselines. Empirical studies has confirmed the better graph learning accuracy results than the existing RL baseline as well as some non-RL structure learning algorithms.

Positive:
1. Formulation of RL order search is interesting.  Authors discussed the advantage of order-based approach over the existing RL Algorithms, which is convincing. The motivation hence is clear.
2. Theoretical justification with Prop 1 justified the said order-based approach.
2. The writing is generally easy to follow. Figure 3 study is appreciated.

Concerns:

1. it is not accurate to state that the exact algorithms can only handle 20.  For example, B&B in (Campos & Ji, 2011) can handle more than 30 nodes, and GOBNILP can regularly handle unto hundreds of variables.
2. Prop 1: since the statement of graphs with the same distribution P(X) is made, is faithfulness assumed?
3.  the output dimension of encoder is d times dimension of s? What is the input dimension of decoder and s hat at each step?
4. Some important ablation study are missing, without them it is hard to judge the exact improvement of each components. For example, what is the performance of the pretrained model when combined with RL-BIC2? Would random sampling order instead of learning it via the proposed approach gives similar performance? How about the orders from other order-based learning approaches (as indicated by authors in the paper)?
5. pre-training model: "consider a scenario where the policy model learned on a data model task is used as the pretrained policy model for other tasks. Such results are demonstrated in Section 5.1" - which part in Section 5.1? how the training datasets are generated? is it generated under the same setting as testing graphs? This seems critical on the evaluation.
6. It could be beneficial that authors show the proposed approach learns a better order than some methods discussed in the paper (Schmidt et al and others). This is important to evaluate the power of RL based approach, since the post processing step is standard.
7. Since the method uses complex structures such as transformer, an efficiency comparison with all methods is warranted. Specially considering many methods can be tuned via hyperparameters to obtain better solutions with longer time (such as hidden dimensions of neural methods, acyclicity threshold from NOTEAR based methods).

---

> ### Author Response · Authors · 2020-11-21
> **Response to Reviewer 3**
>
> We thank the reviewer for valuable comments/suggestions. We provide point by point responses below.
>
> **1. it is not accurate to state that the exact algorithms can only handle 20. For example, [1] can handle more than 30 nodes, and GOBNILP can regularly handle unto hundreds of variables.**
>
> Thanks for this helpful comment. We have updated the statement with a more accurate description based on [1]. To our knowledge, to handle the large problem with hundreds of variables, GLOBNILP also needs to adopt certain approximations like assuming a limited number on the maximum number of parents.
>
> **2. Prop 1: since the statement of graphs with the same distribution $P(X)$ is made, is faithfulness assumed?**
>
> We assume identifiable cases, some of which may implicitly assume faithfulness or other conditions like causal minimality. For example,  Additive Noise Model requires the causal minimality condition, not the faithfulness condition.
>
> **3. the output dimension of encoder is $d$ times dimension of $s$? What is the input dimension of decoder and $s$ hat at each step?**
>
> Yes, the output dimension of encoder is as you said. The input of decoder at $t$-th time step is state $\hat{s}_t$ and $\hat{s}_t\in \mathbb R^{n}$, where $n$ denotes the dimension of $\hat{s}$. We have provided a more detailed description about this  in the Section 4.2 in the revised manuscript. Thanks.
>
> **4. Some important ablation study are missing. For example, what is the performance of the pretrained model when combined with RL-BIC2? Would random sampling order instead of learning it via the proposed approach gives similar performance? How about the orders from other order-based learning approaches (as indicated by authors in the paper)?**
>
> Thanks for this helpful comment that makes our paper more self-contained and informative.
>    1) Our primary goal is to show the pretraining approach is beneficial to the RL based approaches, mainly for accelerating training. We expect that RL-BIC2 could benefit from a pretrained model in terms of training time, but not much in terms of estimate performance. So we did not include its results.
>    2) We have also added the random sampling as a baseline in Tables 1 and 2 in the revised manuscript, which shows a worse performance than CORL.
>    3) We believe that the TPR metric can imply the number of the true edges entailed in a searched ordering, and it can be used to evaluate the quality of an ordering, as we also adopt the same pruning method to a previous ordering-based method. Notice that a pruning method can only remove edges so cannot improve TPR.
>    4) Besides, we have also provided the ablation study about the choice of encoder and decoder in Appendix A.
>
> **5. Regarding the remark: '...which part in Section 5.1? how the training datasets are generated? is it generated under the same setting as testing graphs?...'**
>
> Thanks for this comment that helped us improve the paper's readability.  We have provided a more detailed description on these points in the revised manuscript.
>    1) The experimental results about pretraining method are demonstrated in the Paragraph Pretraining in Section 5.1.
>    2) A detailed description about the dataset generation is provided in Appendix B.
>    3) The testing graphs are different from the pretraining ones in graph sizes, graph degrees, and also the noise types.
>
> **6. It could be beneficial that authors show the proposed approach learns a better order than some methods discussed in the paper. This is important to evaluate the power of RL based approach, since the post processing step is standard.**
>
> Thanks for the suggestion. We believe that the TPR metric can imply the number of the true edges entailed in a searched ordering, and it can be used to evaluate the quality of an ordering. To see this, notice that a pruning method can only remove edges so cannot improve TPR. In our experiments, the pruning method is effective in that it would reduce many false discoveries and only have little effect on true positives if the searched ordering is almost consistent to the underlying graph.
>
> **7. Regarding : '...an efficiency comparison with all methods is warranted...'**
>
> Thanks for this helpful suggestion. We have added a discussion on the computational complexity in Section 4.4 and have reported the total number of episodes and runtime to reach converge on linear Gaussian datasets in Appendix C in the revised manuscript.
> Compared to RL-BIC2, a previous RL-based method, CORL enjoys higher sample efficiency since it converges using lower number of episodes.
> A fact is that the total time of the RL based approach is generally longer than other methods, but we believe the time is acceptable given the gained accuracy.  We are currently working to provide more about running times.
>
> ---
> **References**
>
> [1] Cassio P De Campos and Qiang Ji. Efficient structure learning of bayesian networks using constraints.
> The Journal of Machine Learning Research, 12:663–689, 2011.

---

### Official Review · AnonReviewer1 · 2020-10-29
**Experimental results seem promising, but the method lacks theoretical guarantees of identifiability**

**Rating:** 5
**Confidence:** 4

**Review:**

This paper proposes an RL-based method to learn the causal ordering of variables. Specifically, it formulates the ordering search problem as a Markov decision process, and then uses different reward designs to optimize the ordering generating model. Compared to [1], which uses RL to search in the DAG space, the proposed method is more efficient.

Pros:
1. The presentation is easy to follow.
2. The experimental results seem promising.
3. The idea of searching the causal ordering, instead of in the DAG space,  to improve the efficiency is reasonable.

Cons:
1. However, my main concern is that the proposed method lacks a theoretical guarantee of the identifiability of causal ordering, which is very important in the field of causal discovery.
2. In addition, on linear data, the authors consider large-scale graphs, e.g., the graph which contains 100 nodes. I am wondering why the graph with 100 nodes is not tested on nonlinear data.
3. Furthermore, the authors should also report the time complexity in order to show the improvement of efficiency.

If the authors can solve the above issues, I will increase the score.

Another point is that I am very surprised at the results shown in Table 2. It shows that ICA-LINGAM achieves the true graph even when there are 100 nodes. I am curious about which implementation the authors use.

[1] Shengyu Zhu, Ignavier Ng, and Zhitang Chen. Causal discovery with reinforcement learning. In International Conference on Learning Representations (ICLR), 2020.


Post-rebuttal:
Thanks for the feedback. Some of my concerns have been addressed, but I am still worried about the consistency issue and how to handle the large-scale problem.

---

> ### Author Response · Authors · 2020-11-21
> **Response to Reviewer 1**
>
> We greatly appreciate the reviewer's time and effort. Our detailed response follows.\
> **1. Regarding `my main concern is that the proposed method lacks a theoretical guarantee of the identifiability of causal ordering, which is very important in the field of causal discovery.'**
>
> Thanks for this comment. We agree with the reviewer that identifiability is a key issue in causal discovery. In our opinion, identifiability is only related to the data model (i.e., SEM); for example, if the model follows linear non-Gaussian data, then the true graph is identifiable. In Section 4.2, we shows that given the identifiable model and the causal minimality condition, then the true topological ordering is also identifiable. In our paper, we mostly consider identifiable data models so that it is reasonable to directly compare the estimate with the ground truth.
>
> Perhaps the other important issue is the consistency of the proposed method, that is, whether a method could result in the true graph (or Markov equivalence class) assuming a right score function and certain class of SEMs, when sample number goes to infinity. We have to say that the RL approach can at most lead to local solution so we cannot guarantee the consistency of the method. Nevertheless, compared with the baselines, the empirical results in the finite sample regime is much better.
>
> **2. In addition, on linear data, the authors consider large-scale graphs, e.g., the graph which contains 100 nodes. I am wondering why the graph with 100 nodes is not tested on nonlinear data.**
>
> This is an acute observation. The reason is due to computational costs in rewards: the Gaussian process regression takes much longer time than a linear regression when calculating the BIC score. In our experiments, we make a  time limit for all the experiments, and the 100-node experiments on nonlinear data cannot be finished within the time. So we did not report this result. Comparing with the reward computation time, the RL training takes a much shorter time (see Section 4.4). We believe that our approach is feasible for these cases provided an effective and more efficient score function. Developing such a score function is beyond the scope of the present work and is treated as a future work.
>
> **3. Furthermore, the authors should also report the time complexity in order to show the improvement of efficiency.**
>
> Thanks for this helpful suggestion. We have added a discussion on the computational complexity in Section 4.4 and have reported the total number of episodes to reach converge and runtime on linear Gaussian datasets in Appendix C in the revised manuscript.
> Compared to RL-BIC2, CORL enjoys higher sample efficiency since it converges using lower number of episodes.
>
> **4. Another point is that I am very surprised at the results shown in Table 2. It shows that ICA-LINGAM achieves the true graph even when there are 100 nodes. I am curious about which implementation the authors use.**
>
> We find two Python implementations of ICA-LiNGAM at  https://sites.google.com/site/sshimizu06/lingam and
> https://github.com/cdt15/lingam, respectively. The latter is a Python package that also contains DirectLiNGAM. In our experience, DirectLiNGAM is usually slightly better than ICA-LiNGAM in the same Python package; the implementation of ICA-LiNGAM at  https://sites.google.com/site/sshimizu06/lingam is noticeably better than DirectLiNGAM for relatively large and dense graphs.  We forked the former. Moreover, we increase the default iteration numbers to 20000 and apply the additional thresholding to reduce false discoveries.

---

### Author Response · Authors · 2020-11-20
**General Response**

We appreciate the comments/suggestions from the reviewers that have helped us greatly improve the paper. We have uploaded a revised manuscript, taking into account all the suggestions/comments. Below are some notable changes:
+ By considering R1's suggestion, we provided a discussion on theoretical guarantee of the identifiability of causal ordering in the last two paragraphs of Section 4.2.
+ According to  R1's, R2's and R3's comments, we added a discussion on the computational complexity (Section 4.4) and reported some running times and the total number of episodes before RL converges on linear Gaussian datasets (Appendix C).
+ We provided some ablation study about the choice of encoder and decoder module (Appendix A) and provided a more detailed description about the  generation of synthetic data (Appendix B.2), according to R2's and R3's comments.
+ Following R2's suggestion, we provided more diverse sets of experiment. Specifically, these settings vary along five dimensions: level of edge sparsity (2d and 5d), graph type (ER and SF), number of nodes (30, 50 and 100), causal functions (linear Gaussian, LiNGAM and Gaussian process) and sample size (Please see Appendix F).
+ We provided a more detailed explanation about the RL formulation in Paragraph Encoder and Paragraph Decoder (Section 4.2), according to R2's comments.
+ We gave a discussion on some more recently ordering-based works such as Sparest Permutation algorithm  and Greedy Sparest Permutation algorithm (Section 2), highlighted the contribution of our work and improved some inappropriate expressions, in response to R3's and R4's comments.

We once again thank all the reviewers for the effort they put into reviewing our submission.

---

### Decision · Program_Chairs · 2021-01-07
**Final Decision**

**Decision:**

Reject

**Comment:**

In this paper, the authors propose an RL-based method for learning DAGs based on searching over causal orders instead of graphs. Order search for learning DAGs is a well-studied problem, and it is well-known that this can relieve some of the burden of searching through the space of DAGs. Several reviewers raised legitimate concerns regarding the experiments,  and without identifiability or theoretical results to advance the state of the art, the contribution of this work is limited.